# ProDyG: Progressive Dynamic Scene Reconstruction via Gaussian Splatting from Monocular Videos

**Shi Chen**
ETH Zürich

**Erik Sandström**
Google

**Sandro Lombardi**
Independent Researcher

**Siyuan Li**
ETH Zürich

**Martin R. Oswald**
University of Amsterdam

## Abstract

Achieving truly practical dynamic 3D reconstruction requires online operation, global pose and map consistency, detailed appearance modeling, and the flexibility to handle both RGB and RGB-D inputs. However, existing SLAM methods typically merely remove the dynamic parts or require RGB-D input, while offline methods are not scalable to long video sequences, and current transformer-based feedforward methods lack global consistency and appearance details. To this end, we achieve online dynamic scene reconstruction by disentangling the static and dynamic parts within a SLAM system. The poses are tracked robustly with a novel motion masking strategy, and dynamic parts are reconstructed leveraging a progressive adaptation of a Motion Scaffolds graph. Our method yields novel view renderings competitive to offline methods and achieves on-par tracking with state-of-the-art dynamic SLAM methods. Project page: cs-vision.github.io/ProDyG.github.io

## 1 Introduction

Dynamic scene reconstruction is fundamental to problems like action recognition, scene understanding, autonomous driving, robotics and augmented reality, because it provides a temporally consistent spatial understanding of how objects and agents move and interact in their environment — an essential prerequisite for any system to perceive, predict, and act in the world. This problem has been tackled in various ways *e.g.* as an online SLAM task [4, 89, 34], an offline reconstruction task [29, 73, 79, 39, 56, 55], and lately as a feedforward task [86, 87, 11, 69]. However, most dynamic SLAM works ignore the dynamic parts [89, 4], only reconstructing the static world, only track rigid objects [85, 43] or are restricted to object-centric reconstruction [45, 58]. Offline methods typically separate the reconstruction task into pose estimation followed by reconstruction [55, 56], or are not scalable to long input videos due to their reliance on global optimization over all past frames [29, 73]. Feed-forward methods train large transformers for online dynamic scene reconstruction, but are yet to achieve global pose consistency and only produce point clouds [69, 11, 86].

Despite recent progress, existing methods fall short of at least one of the requirements for practical dynamic scene reconstruction: (1) **online operation**, tightly coupling pose estimation and dense map reconstruction for scalability; (2) **global pose and map consistency**; (3) **expressive representations** like 3DGS [23] for detailed appearance and geometry; and (4) **flexibility** to handle both **RGB and RGB-D** input. SLAM systems often ignore dynamics or lack detail; feedforward methods trade consistency and accuracy for speed; and offline methods are challenging to scale.

We propose *ProDyG*[1], a method for **Pro**gressive, **Dy**namic scene reconstruction with **G**aussians from monocular input, that meets all four criteria. Our contributions are:

---

[1] Pronounced "Prodigy".

39th Conference on Neural Information Processing Systems (NeurIPS 2025).

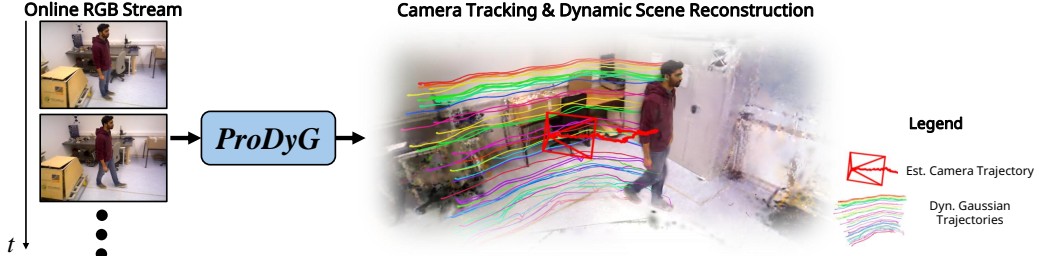

Figure 1: **ProDyG:** Given an online RGB stream, ProDyG robustly tracks the camera and progressively reconstructs the static background and the dynamic foreground using 3D Gaussian Splatting [23] (3DGS). ProDyG reconstructs dynamic scenes with high-quality novel view synthesis, by fusing observations from different timestamps into a consistent dynamic 3DGS [23] representation with Motion Scaffolds [29]. On the right, we show a rendering from a novel view overlaid with the estimated camera trajectory and dynamic Gaussian trajectories.

- A motion mask prediction strategy, using dynamic flow and refined with semantic point prompting [50]. This is integrated into the SLAM backend for robust pose estimation.

- An **online** dynamic reconstruction pipeline that uses Motion Scaffolds [29] to propagate 3DGS in space and time.

- Support for both RGB and RGB-D, with ProDyG as the **first online RGB-only** method.

- Competitive novel view synthesis results against state-of-the-art offline methods.

## 2 Related Work

**Dynamic Dense SLAM.** Dynamic dense SLAM methods mainly address two scenarios: filtering dynamic content to reconstruct only static environments [4, 13, 90, 89, 77, 19, 15, 81, 44, 5, 74, 59, 48, 57, 82, 37, 32, 88, 1, 24, 7, 64, 26, 72, 51], or explicitly tracking and reconstructing dynamic objects alongside static structures [85, 3, 43, 17, 52, 53, 78, 14, 45, 58, 8, 65, 34, 35, 31]. Basic strategies for handling dynamic elements include outlier filtering, robust loss functions [90, 44, 5], or covariance scaling [1], but these fail with extensive dynamic content. Instead, a motion mask is estimated with optical flow [88, 82], semantic segmentation [26, 14, 35, 57, 4, 33, 77, 51, 81, 74, 85, 59, 72, 53] or a combination of the two [34, 19, 15, 52], with uncertainty-guided loss functions [89, 37], with motion segmentation networks [57], point tracking [65], unsupervised clustering [17] or via conditional random fields [32]. In many real-world scenarios, reconstructing dynamic objects is critical. Some traditional works track rigid dynamic content during SLAM [85, 3, 43, 17, 52, 53, 78, 14] or perform object-centric non-rigid reconstruction [45, 58, 8]. While the above-mentioned methods can reconstruct dynamic point clouds [4, 85, 74, 59, 88, 17], surfels [52, 53], oct-trees [81] or signed distance functions [48, 45, 58, 8], they struggle to model photometric details and lighting effects needed for scene understanding and photorealistic rendering. In response, neural implicit methods [33, 77, 51, 19] have been proposed, but they are too slow for real-time SLAM. Dynamic 3DGS [23] SLAM has emerged as a solution [15, 34, 35, 31, 89, 32, 72, 26] with concurrent works such as DynaGSLAM [34] using dynamic scene flow from optical flow to update the Gaussian means, [65] adapts [79] to the online setting and [35] uses an MLP to deform the dynamic parts. These works cannot handle pure RGB input. For a recent survey on dynamic SLAM, we refer to [71].

**Dynamic 3D Reconstruction.** Dynamic 3D reconstruction shares many similarities with dynamic SLAM, but videos are processed offline [29, 73, 79, 9, 21, 80, 27, 28, 36, 40, 41, 49, 60, 84, 2, 16, 39, 25, 91, 18, 38, 68]. These works typically implement a strategy for deforming dynamic content over time. Some optimize a deformation field via an MLP [2, 60, 40, 80, 25, 73], use motion basis functions [16, 49, 68, 36, 29, 28, 27, 21, 39, 18], or extend 3D Gaussian Splatting [23] with a time attribute [79, 9]. Due to the ill-posed nature of dynamic reconstruction, strong priors are commonly used, such as point trackers, optical flow, and regularizers like as-rigid-as-possible (ARAP), constant velocity, and acceleration [29, 41, 56, 68, 39]. Inductive biases also help, such as MLP smoothness

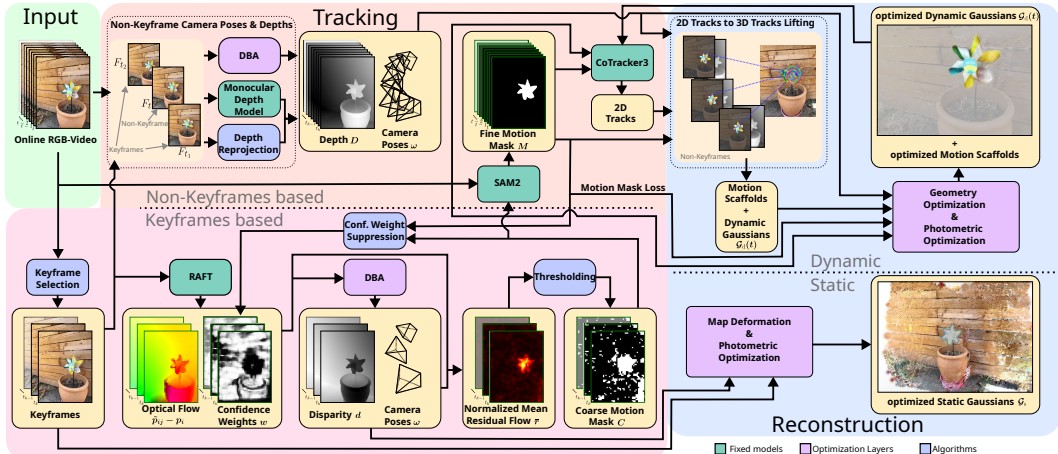

Figure 2: **ProDyG Architecture.** We achieve motion-agnostic online tracking by leveraging [54] to first create keyframe-based coarse motion masks, from which we seed prompts for SAM2 to distill per-frame fine-grained masks. ProDyG processes batches of frames incrementally, employing a keyframe selection similar to [54]. Static background is reconstructed by optimizing the static set of Gaussians with proxy depth maps [54]. For dynamic reconstruction, we attach Gaussians to Motion Scaffolds [29], which are initialized by lifting 2D tracks to 3D, to encode a dense motion field. Subsequent to a final geometric and photometric optimization, the Motion Scaffolds and dynamic Gaussians are extended temporally when a new batch of images arrives.

and continuity, or selecting a sparse set of motion bases. While some of these works use SLAM for pose estimation, reconstruction is a post-processing step [55, 56].

**Feed-forward Methods.** Feed-forward methods inspired by DUSt3R [70] and MASt3R [30] have been proposed to solve dynamic 3D reconstruction. These works rely on large-scale training of transformer networks and can infer pointmaps from dynamic input image pairs [86, 87, 11, 13, 6, 69]. However, they typically entangle static and dynamic points [69, 86], meaning that the motion of dynamic points cannot be tracked over time. In concurrent work, [11, 13, 87] propose to enable 3D correspondence estimation of dynamic 3D points. DAS3R [75] learns to predict motion masks building on MonST3R [86], and only predicts the static world, using 3DGS [23]. Common to all feed-forward methods is that they process the video either in image pairs or in a sliding window, without guaranteeing global pose and map consistency, contrary to SLAM. In contrast, we perform globally consistent and online dynamic reconstruction using SLAM.

## 3 Method

ProDyG is an online dynamic dense mapping and tracking system that robustly tracks a monocular camera (sec. 3.1) while also reconstructing and disentangling the static and dynamic parts (sec. 3.3) with a 3D Gaussian Splatting representation (sec. 3.2). For an overview, see fig. 2.

### 3.1 Motion-Agnostic Online Camera Tracking

**Flow-Based Robust Camera Tracking.** We employ Splat-SLAM [54] as our tracking backend, *i.e.* we maintain a factor graph storing camera extrinsics $\omega_i$, disparity estimates $d_i \in \mathbb{R}^{(H \times W) \times 1}$ per keyframe (node) $i$, and optical flow $\tilde{p}_{ij} - p_i \in \mathbb{R}^{(H \times W \times 2) \times 1}$ per edge $(i, j)$, where $p_i$ is the flattened pixel grid from keyframe $i$, and $\tilde{p}_{ij}$ represents the flattened predicted pixel coordinates when $p_i$ is projected into keyframe $j$ using optical flow. We also store the confidence $w_{ij} \in \mathbb{R}^{(H \times W) \times 1}$ associated with the optical flow. Tracking is achieved following Dense Bundle Adjustment (DBA) [66], where the keyframe poses $\omega$ and disparities $d$ are optimized with a reprojection error:

$$\underset{\omega, d}{\arg\min} \sum_{(i,j) \in \mathcal{E}} \|\tilde{p}_{ij} - p_{ij}\|_{\Sigma'_{ij}}^2, \quad p_{ij} = K\omega_j^{-1}(\omega_i(1/d_i)K^{-1}[p_i, 1]^T), \quad \Sigma'_{ij} = \text{diag}(w_{ij}\bar{C}_i). \quad (1)$$

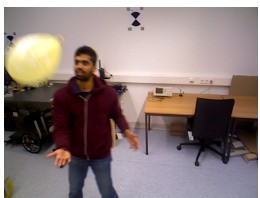 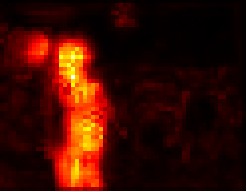 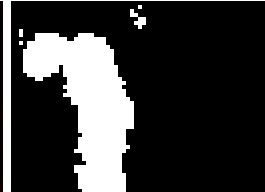 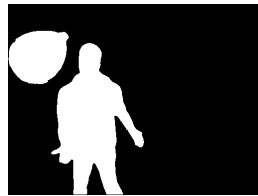

(a) Input Image   (b) Mean Mag. $\bar{r}_i$ of Residual Flow   (c) Coarse Motion Mask   (d) Fine Motion Mask

Figure 3: **Semantic-guided Motion Mask Refinement.** The flow magnitude $\bar{r}_i$ (b) is thresholded to yield (c). Point prompts from (c) are used as input to SAM2 [50] to yield the fine motion masks (d).

Here, $\mathcal{E}$ are the edges of a local factor graph, applied in a sliding window manner, $K$ is the camera intrinsics, $\|\cdot\|_{\Sigma'_{ij}}$ denotes the Mahalanobis distance with confidence weights suppressed by a coarse binary motion mask $C_i$, defined as 1 where motion is detected and $\bar{C}_i$ stands for negation of $C_i$.

We generate the coarse motion masks $\{C_i\}$ as follows. After each DBA iteration, we compute the residual flow $\hat{r}_{ij} = (\tilde{p}_{ij} - p_i) - (p_{ij} - p_i) = \tilde{p}_{ij} - p_{ij}$ by subtracting the camera-induced flow $p_{ij} - p_i$ from the estimated optical flow $\tilde{p}_{ij} - p_i$. This residual is near zero in static regions and larger in dynamic ones. Dynamic areas are identified by evaluating the normalized mean magnitude $\bar{r}_i(x,y)$ of $\hat{r}_{ij}$ over the connected target keyframes $N_i$ for each keyframe $i$

$$\bar{r}_i(x,y) = \frac{1}{|N_i|} \sum_{j \in N_i} \frac{\|\hat{r}_{ij}(x,y)\|}{\frac{1}{HW}\sum_{x',y'} \|\tilde{p}_{ij}(x',y') - p_i(x',y')\|} \ . \tag{2}$$

For each keyframe, $C_i$ is computed by thresholding $\bar{r}_i(x,y)$ at the top 20%, assigning zero weight to potentially dynamic regions during DBA. As $\bar{r}_i(x,y)$ is updated each DBA iteration, it is progressively refined, enabling robust pose estimation in dynamic environments.

**Semantic-Guided Motion Mask Refinement.** To improve the separation of dynamic and static regions for reconstruction (sec. 3.3), we use SAM2 [50] to generate fine-grained motion masks for every frame. We show an example visualization of the input image, the mean magnitude $\bar{r}_i$ of the residual flow, the coarse motion mask, and the fine motion mask in fig. 3.

Our approach has two phases: initialization and incremental prediction. During initialization, we generate fine motion masks by prompting SAM2 at the centroids of connected regions in the median-filtered coarse masks $C_i$. During incremental prediction, we extend existing object-wise masks into new frames and add new segmentations based on the coarse masks. We validate each segmentation by counting prompt point candidates to filter false positives. Before this phase, we adjust the threshold (top 20% at initialization) based on the dynamic pixel ratio in the latest fine motion mask, adapting to sequence-specific motion. Detailed algorithms are in the supplemental material.

With the fine motion masks $\{M_i\}$, we further enhance our tracking robustness by replacing the coarse mask $C_i$ with the fine during DBA (eq. (1)) as $\Sigma'_{ij} = \text{diag}(w_{ij}(1 - M_i))$. This semantic-guided approach provides more accurate exclusion of dynamic distractors for subsequent iterations of DBA, including local BA, global BA and loop BA [54].

## 3.2   Static and Dynamic Map Representation

We represent the scene map $\mathcal{G} = \{\mathcal{G}_s, \mathcal{G}_d(t)\}$ as a set of 3D Gaussians [23], split into static $\mathcal{G}_s = \{g_i\}_{i=1}^{N_s}$ and time-varying dynamic components $\mathcal{G}_d(t) = \{g_i(t)\}_{i=1}^{N_d}$. Each Gaussian—whether static or dynamic—is parameterized by a mean $\boldsymbol{\mu}_i \in \mathbb{R}^3$, rotation matrix $R_i \in \mathbb{R}^{3\times3}$, scale $\mathbf{s}_i \in \mathbb{R}^3$, opacity $o_i \in [0,1]$, and RGB color $\mathbf{c}_i \in \mathbb{R}^3$. Rendering is performed by first projecting the 3D Gaussians onto the image plane, approximated as 2D Gaussians. The 2D opacity at a pixel $p$ is then

$$\alpha_i(p) = o_i \exp\left(-\frac{1}{2}(p - \boldsymbol{\mu}'_i)^\top \Sigma_i'^{-1}(p - \boldsymbol{\mu}'_i)\right) \ , \tag{3}$$

where $\boldsymbol{\mu}'_i = K\omega^{-1}\boldsymbol{\mu}_i$ with $\omega$ the camera-to-world pose, $K$ the camera intrinsics, and $\Sigma'_i = P\Sigma_i P^\top = P(R_i S_i S_i^\top R_i^\top)P^\top$, with $S_i = \text{diag}(\mathbf{s}_i)$ and $P$ the affine projection [92]. Given the pose

$\omega$, we render a pixel $p$ following the 3DGS [23] pipeline for RGB images $C$, RaDe-GS [83] for depth maps $D^r$, and additionally compute opacity maps $O$ as

$$C(p) = \sum_{i \in \mathcal{K}} \mathbf{c}_i \alpha_i \prod_{j=1}^{i-1}(1-\alpha_j), \quad D^r(p) = \sum_{i \in \mathcal{K}} \hat{d}_i \alpha_i \prod_{j=1}^{i-1}(1-\alpha_j), \quad O(p) = \sum_{i \in \mathcal{K}} \alpha_i \prod_{j=1}^{i-1}(1-\alpha_j). \quad (4)$$

Here, $\mathcal{K}$ is the set of Gaussians projected to $p$. We use the unbiased RaDe-GS rasterizer, which replaces the z-buffer depth with $\hat{d}_i$, the ray–Gaussian intersection depth along the viewing ray.

### 3.3 Progressive Dynamic Scene Reconstruction

We propose a progressive approach to dynamic scene reconstruction, which enables online processing and allows the system to adapt to the continuously evolving scene geometry and camera motion. ProDyG uses the same keyframe selection strategy as in Splat-SLAM [54], and the addition of new keyframes triggers progressive reconstruction of both static and dynamic regions. To achieve global map consistency, we apply 3DGS map deformations for static regions followed by photometric optimization as in [54]. For dynamic Gaussians, we extend the Motion Scaffold representation [29] to handle incremental updates. Next, we review the necessary steps.

**Non-Keyframe Camera Poses and Depths.** Dynamic scene reconstruction benefits from temporally dense supervision to capture smooth motion, yet the SLAM backend [54] operates on sparse keyframes for efficiency. To bridge this gap, we derive the camera poses and depth maps for non-keyframes that exist between keyframes.

To obtain non-keyframe camera poses, we follow the practice of DROID-SLAM [66]. For each non-keyframe $F_t$ positioned between two neighboring keyframes $F_{t_1}$ and $F_{t_2}$, we construct a temporary local graph with edges connecting both keyframes to $F_t$. We then execute the recurrent update operator and the Dense Bundle Adjustment (DBA) layer on this graph, which optimizes the non-keyframe poses based on the already optimized keyframe poses.

For non-keyframe depth estimation, we leverage reprojection from neighboring keyframes. Given a non-keyframe $F_t$ with its derived camera-to-world pose $\omega_t$ and neighboring keyframes $F_{t_1}$ and $F_{t_2}$ with poses $\omega_{t_1}$ and $\omega_{t_2}$, we first back-project the pixels of both keyframes into 3D:

$$p_{t_1}^{3D}(p_{t_1}) = \omega_{t_1}(1/d_{t_1})K^{-1}[p_{t_1}, 1]^T \;\; , \;\; p_{t_2}^{3D}(p_{t_2}) = \omega_{t_2}(1/d_{t_2})K^{-1}[p_{t_2}, 1]^T \;\; , \quad (5)$$

where $K$ is the camera intrinsic matrix, and $d_{t_1}$ and $d_{t_2}$ are the disparities of keyframes $F_{t_1}$ and $F_{t_2}$, respectively. We then reproject all back-projected 3D points $p_{t_1,t_2}^{3D} = p_{t_1}^{3D} \cup p_{t_2}^{3D}$ from both keyframes to the camera of the non-keyframe $F_t$ and record their distance to the camera along the z-axis:

$$p_t^{3D} = \omega_t^{-1} p_{t_1,t_2}^{3D} \;\; , \;\; p_t = K p_t^{3D} \;\; , \;\; D_t^{repro}(p_t) = (p_t^{3D})_z \;\; . \quad (6)$$

We filter out points that are back-projected from dynamic pixels or using invalid disparity estimates (determined by a multi-view consistency check as in [54]). Using the remaining reprojected 2D coordinates and their corresponding depth values, we bilinearly interpolate to create a reprojection depth map $D_t^{repro}$. Finally, to get dense depth, we align a monocular depth estimate $D_t^{mono}$ with $D_t^{repro}$ by estimating scale $\theta_t$ and shift $\gamma_t$ parameters through least squares fitting:

$$\theta_t, \gamma_t = \arg\min_{\theta,\gamma} \sum_{(u,v)} \left( (\theta D_t^{mono}(u,v) + \gamma) - D_t^{repro}(u,v) \right)^2 \;\; , \;\; D_t^{aligned} = \theta_t D_t^{mono} + \gamma_t \;\; . \quad (7)$$

**Motion Scaffolds.** We adopt Motion Scaffolds [29] (MoSca) as our dynamic representation. MoSca a structured graph $(\mathcal{V}_d, \mathcal{E}_d)$ representing the underlying motion of the scene, where each node $v^{(m)} \in \mathcal{V}_d$ encodes the motion trajectory of a specific region. Each MoSca node $v^{(m)}$ is defined as

$$v^{(m)} = ([\mathbf{Q}_1^{(m)}, \mathbf{Q}_2^{(m)}, \ldots, \mathbf{Q}_T^{(m)}], r^{(m)}) \;\; , \quad (8)$$

where $\mathbf{Q}_t^{(m)} \in SE(3)$ represents the per-timestep rigid transformation at time $t$, and $r^{(m)}$ is an RBF radius parameterizing the node's influence. These nodes are initially anchored by lifting 2D pixel trajectories using the estimated camera poses $\omega_t$ and depth maps $D_t^{aligned}$ computed in eq. (7).

Similar to the static map deformation [54], we also deform the dynamic Gaussians to reflect the continuously updated pose and depth estimates from the SLAM backend. Each dynamic Gaussian is defined relative to a MoSca node (not in world coordinates) and by re-anchoring the MoSca nodes, the dynamic Gaussians are updated accordingly. The MoSca nodes are re-anchored in 3D when new pose and depth updates are available. Thus, ProDyG achieves global dynamic map consistency prior to optimization. This ensures better convergence and consistency between the static and dynamic components of the scene.

Following MoSca [29], the set of dynamic Gaussians $\mathcal{G}_d(t)$ at any query timestamp $t$ is formed by warping each individual Gaussian $g_i(t_i^{\mathrm{ref}}) = (\mu_i, R_i, \mathbf{s}_i, o_i, \mathbf{c}_i; t_i^{\mathrm{ref}}, \Delta\mathbf{w}_i)$ from its reference timestamp $t_i^{\mathrm{ref}}$ (the timestamp where it is initialized) to $t$

$$\mathcal{G}_d(t) = \{(\mathbf{T}_i(t)\mu_i, \mathbf{T}_i(t)R_i, s_i, o_i, \mathbf{c}_i \mid \mathbf{T}_i(t) = \mathcal{W}(\mu_i, \mathbf{w}(\mu_i) + \Delta\mathbf{w}_i; t_i^{\mathrm{ref}}, t)\}_{i=1}^{N_d} \quad, \qquad (9)$$

Here, $\mathbf{w}(\cdot)$ is the base RBF skinning weight parametrized by $\{r^{(m)}\}_{m \in \mathcal{E}_d(m^*)}$, where the neighborhood $\mathcal{E}_d(m^*)$ consists of the nearest MoSca node $v^{(m^*)}$ and all nodes connected to $v^{(m^*)}$, and $\Delta\mathbf{w}_i$ are learnable per-Gaussian skinning weight corrections. The warping function $\mathcal{W}(\cdot)$ is computed using Dual Quaternion Blending (DQB) [22]:

$$\mathcal{W}(\mathbf{x}, \mathbf{w}; t_{src}, t_{dst}) = \mathrm{DQB}\left(\{w_m, \Delta\mathbf{Q}^{(m)}\}_{m \in \mathcal{E}_d(m^*)}\right) \quad, \qquad (10)$$

where $\Delta\mathbf{Q}^{(m)} = \mathbf{Q}_{t_{dst}}^{(m)}(\mathbf{Q}_{t_{src}}^{(m)})^{-1}$ is the relative transformation between $t_{src}$ and $t_{dst}$ for node $m$.

**Progressive Construction of Motion Scaffolds.** After the initial bootstrapping phase of the SLAM backend, we initialize the Motion Scaffolds and dynamic Gaussians. First, we employ CoTracker3 [20] to generate dense long-term 2D pixel trajectories within the fine motion masks $\{M_i\}$ (sec. 3.1) and corresponding per-timestamp visibility labels. Similar to [29], these 2D trajectories are first lifted into 3D space at visible timestamps using the camera poses and depth maps estimated by the backend, while we linearly interpolate between nearby observations at invisible timestamps. Finally, we sample a subset of the lifted 3D tracks to serve as the initial positions for the MoSca nodes.

Subsequently, we carry out geometry optimization on the initialized MoSca nodes as described in [29] to infer rotations and positions of invisible nodes, minimizing the as-rigid-as-possible (ARAP) loss, velocity consistency loss and acceleration consistency loss.

After the geometry optimization of the Motion Scaffolds, we initialize the dynamic Gaussians at 3D positions obtained by back-projecting pixels within the fine motion masks. Finally, we perform a photometric optimization of both the Motion Scaffolds and the dynamic Gaussians using a combination of losses following [29]: an RGB loss $\mathcal{L}_{\mathrm{rgb}}$ that enforces color consistency, a depth loss $\mathcal{L}_{\mathrm{depth}}$ that aligns rendered depth $D^{\mathrm{r}}$ with the aligned monocular depth $D^{\mathrm{aligned}}$ (eq. (7)), a track loss $\mathcal{L}_{\mathrm{track}}$ that ensures consistent motion with the 2D trajectories, and the aforementioned ARAP, velocity, and acceleration losses for geometric regularization. In addition to [29], we introduce a novel motion mask loss that penalizes the rendered opacity of dynamic Gaussians at pixels identified as static by the motion masks

$$\mathcal{L}_{\mathrm{mask}} = \frac{1}{|\mathcal{P}_t^{\mathrm{static}}|} \sum_{p \in \mathcal{P}_t^{\mathrm{static}}} O(p) \quad, \qquad (11)$$

where $\mathcal{P}_t^{\mathrm{static}}$ is the set of static pixels at timestamp $t$ given by negating the motion mask $M_t$, and $O(p)$ is the accumulated opacity of dynamic Gaussians at pixel $p$ (eq. (4)). This loss effectively prevents dynamic Gaussians from overflowing into static regions, maintaining a clean separation between static and dynamic components of the scene.

As more frames become available, we extend our dynamic reconstruction. For each new frame batch, we run CoTracker3 [20] in a temporal window of the new frame batch and an 8-frame overlap with previously processed frames to ensure continuity in the reconstruction. First, we identify "recently visible" tracks as those marked as visible for at least 4 frames within the 8 overlapping frames and extend them into the new frame batch. These extended 2D tracks are then lifted into 3D using the latest camera pose and depth estimates. To identify newly visible dynamic regions, we back-project

all pixels within the fine motion masks of the new frames into 3D and perform a spherical search to determine whether each back-projected point has at least one lifted 3D track within a predefined radius $r_{\text{search}}$. Pixels without nearby 3D tracks are marked as "newly-seen", representing previously unobserved portions of dynamic objects. To capture these newly-seen regions, we execute a second run of the point tracker, specifically querying newly-seen pixels. Finally, we run a third tracking pass within the entire dynamic regions of the new frames to replenish the density of visible 2D tracks. This multi-stage tracking strategy ensures dense coverage of all dynamic elements in the scene.

Within the new temporal window, all extended and newly added 2D tracks are lifted into 3D using the same procedure as during initialization. For temporal consistency, we warp new tracks to past timestamps with DQB (Eqn.10), using Motion Scaffolds from the previous update. This leverages the latest photometric optimization to guide newly-seen 3D tracks through past frames despite invisibility. We initialize dynamic Gaussians only at newly-seen pixels, then jointly optimize geometry and appearance over the expanded MoSca and dynamic Gaussians using the same losses as during initialization.

## 4 Experiments

### 4.1 Experimental Setup

**Implementation Details.** All experiments were conducted on a cluster with an AMD EPYC 7H12 or 7742 CPU and an NVIDIA A6000 GPU. The kernel size of the median filter used to denoise the coarse motion masks is $5 \times 5$. The spherical search radius for "newly-seen" pixel identification is $r_{\text{search}} = 0.02m$. For geometry and photometric optimization, we keep our loss weights identical with those applied in MoSca [29], and set $\lambda_{\text{mask}} = 1$ as the weight of $\mathcal{L}_{\text{mask}}$. For more implementation details, we refer to the supplemental material.

**Datasets.** We evaluate our camera tracking on the Bonn RGB-D Dynamic Dataset [47] and the TUM RGB-D Dataset [62] (dynamic scenes). Since existing works report tracking results on different sets of sequences, we select four mostly used sequences from each dataset to evaluate our method. For rendering, we report novel view synthesis (NVS) results both qualitatively and quantitatively on the iPhone dataset [12]. To align with Shape of Motion [68], we evaluate our method and all baselines on the 5 sequences used in [68] with the 2x downsampled image resolution. For a fair comparison, we use the preprocessed motion masks given by [68] for all our experiments on the iPhone Dataset.

**Baselines.** For tracking, we compare with various works on RGB and RGB-D SLAM. The main baseline is Splat-SLAM [54] since we base our tracking pipeline on it. For rendering, the baseline methods are NVS-capable monocular Gaussian-based dynamic reconstruction methods including Shape of Motion [68], DynOMo [56], MoSca [29] and Gaussian Marbles [61].

**Metrics.** For tracking, we evaluate ATE RMSE [$cm$] [63] after aligning the estimated camera trajectory with the ground truth via Umeyama alignment [67]. For NVS, we report PSNR, SSIM and LPIPS evaluated within the covisibility masks provided by [12] and averaged over all novel views.

### 4.2 Tracking

In tab. 1, we evaluate tracking performance on the Bonn RGB-D Dynamic Dataset [47] and the TUM RGB-D Dataset [62]. For both datasets, ProDyG performs competitively among all the RGB-D and RGB SLAM works and shows a significant advantage over the main baseline Splat-SLAM [54]. This improvement validates the effectiveness of our motion-agnostic camera tracking method introduced in sec. 3.1. WildGS-SLAM [89] is the only baseline method to outperform ProDyG on both datasets. Since both methods build upon the tracking framework of Splat-SLAM [54], we attribute this performance gap to two key factors: (1) WildGS-SLAM employs a test-time-optimized MLP that produces soft uncertainty masks to suppress confidence weights. As it reconstructs only the static background, it can aggressively suppress regions beyond actual dynamic object boundaries (*e.g.* moving shadows) without degrading mapping quality. In contrast, ProDyG prioritizes accurate dynamic reconstruction and therefore requires motion masks with precise boundaries, which may miss some effective distractors. (2) WildGS-SLAM benefits from additional DINOv2 [46] features and on-the-fly training of the uncertainty MLP at the cost of higher computational complexity, while the primary computational overhead of our tracking approach comes from SAM2 [50] inference.

| Method | Type | Bonn RGB-D Dynamic Dataset [47] | | | | | TUM RGB-D Dataset [62] | | | | |
|---|---|---|---|---|---|---|---|---|---|---|---|
| | | Ball | Ball2 | Pers | Pers2 | Avg. | f3/ws | f3/wx | f3/wr | f3/whs | Avg. |
| *RGB-D Input* | | | | | | | | | | | |
| ORB-SLAM2 [44] | S | 6.5 | 23.0 | 6.9 | 7.9 | 11.1 | 40.8 | 72.2 | 80.5 | 72.3 | 66.45 |
| NICE-SLAM [90] | S | 24.4 | 20.2 | 24.5 | 53.6 | 30.7 | 79.8 | 86.5 | 244.0 | 152.0 | 140.57 |
| ReFusion [48] | R | 17.5 | 25.4 | 28.9 | 46.3 | 29.5 | 1.7 | 9.9 | 40.6 | 10.4 | 15.7 |
| DynaSLAM (N+G) [4] | R | 3.0 | 2.9 | 6.1 | 7.8 | 5.0 | 0.6 | 1.5 | 3.5 | 2.5 | 2.03 |
| DG-SLAM [76] | R | 3.7 | 4.1 | 4.5 | 6.9 | 4.8 | 0.6 | 1.6 | 4.3 | - | - |
| RoDyn-SLAM [19] | R | 7.9 | 11.5 | 14.5 | 13.8 | 11.9 | 1.7 | 8.3 | - | 5.6 | - |
| DDN-SLAM (RGB-D) [33] | R | 1.8 | 4.1 | 4.3 | 3.8 | 3.5 | 1.0 | 1.4 | 3.9 | 2.3 | 2.15 |
| *RGB Input* | | | | | | | | | | | |
| DSO [10] | S | 7.3 | 21.8 | 30.6 | 26.5 | 21.6 | 1.5 | 12.9 | 13.8 | 40.7 | 17.23 |
| DROID-SLAM [66] | S | 7.5 | 4.1 | 4.3 | 5.4 | 5.3 | 1.2 | 1.6 | 4.0 | 2.2 | 2.25 |
| MonoGS [42] | S | 15.3 | 17.3 | 26.4 | 35.2 | 23.6 | 1.1 | 21.5 | 17.4 | 44.2 | 21.05 |
| Splat-SLAM [54] | S | 8.8 | 3.0 | 4.9 | 25.8 | 10.6 | 2.3 | 1.3 | 3.9 | 2.2 | 2.43 |
| DDN-SLAM (RGB) [33] | R | - | - | - | - | - | 2.5 | 2.8 | 8.9 | 4.1 | 4.58 |
| MegaSaM [37] | R | 3.7 | 2.6 | 4.1 | 4.0 | 3.6 | 0.6 | 1.5 | 2.6 | 1.8 | 1.63 |
| WildGS-SLAM [89] | R | 2.7 | 2.4 | 3.6 | 3.1 | 2.94 | 0.4 | 1.3 | 3.3 | 1.6 | 1.63 |
| DynaMoN (MS) [55] | D | 6.8 | 3.8 | 2.4 | 3.5 | 4.1 | 1.4 | 1.4 | 3.9 | 2.0 | 2.18 |
| DynaMoN (MS&SS) [55] | D | 2.8 | 2.7 | 14.8 | 2.2 | 5.6 | 0.7 | 1.4 | 3.9 | 1.9 | 1.98 |
| D4DGS-SLAM* [65] | D | 3.6 | 3.9 | 4.5 | 5.2 | 4.3 | - | - | - | - | - |
| 4D-GS SLAM* [35] | D | 2.4 | 3.7 | 8.9 | 9.4 | 6.1 | 0.5 | 2.1 | 2.6 | - | - |
| **ProDyG (Ours)** | D | 2.7 | 2.6 | 4.9 | 2.9 | 3.29 | 1.6 | 1.2 | 3.0 | 1.7 | 1.89 |

Table 1: **Tracking Performance on Bonn RGB-D Dynamic Dataset [47] and TUM RGB-D Dataset [62].** (ATE RMSE ↓ [cm]). Best results are highlighted as first , second , third and concurrent works with*. We take the numbers from [89] except [89, 65, 35]. We categorize each method into static reconstruction (S), robust against dynamics (R) and producing a globally consistent dynamic model (D). ProDyG is competitive with WildGS-SLAM [89] and MegaSaM [37] while explicitly reconstructing a consistent dynamic model, contrary to [89, 37].

## 4.3 Rendering

In tab. 2, we evaluate NVS performance quantitatively on the iPhone Dataset [12]. Notably, we differentiate the experiment setups of each methods using the checkboxes, whether the reconstruction or tracking is executed online, and whether the input modalities are RGB only or RGB-D. As shown in tab. 2, ProDyG outperforms the offline method Shape of Motion [68] in both PSNR and SSIM, while running both the tracking and dynamic scene reconstruction online. This is a significantly more difficult task. Compared to MoSca [29], which is an offline method that shares the same representation of the motion field with our framework, ProDyG shows only minimal disadvantages when optimized with RGB-D input. This demonstrates the effectiveness of our progressive dynamic reconstruction. DynOMo [56] is the only method capable of online reconstruction with precomputed camera poses, and ProDyG shows a significant advantage over DynOMo when tested under the same constraint. Furthermore, ProDyG still maintains reasonably good performance when only having access to RGB images, while none of the baseline methods is capable of operating under the same setting. Finally, we evaluate NVS when estimated poses from the SLAM backend are loaded offline to the online mapper. The performance is very similar in both RGB-D and RGB-only modes, showing that our method can find the offline solution, even when optimized in pure online mode.

In fig. 4, we show qualitative comparisons of novel view renderings on the iPhone Dataset [12]. Compared to other methods, DynOMo [56] exhibits worse quality in novel view renderings due to lack of motion constraints (*e.g.* 2D point tracking). Therefore, ProDyG is essentially the first method to support high-quality novel view synthesis of dynamic scenes through online reconstruction from monocular videos. Due to the motion mask loss (eq. (11)), dynamic objects reconstructed by our method tend to show more accurate silhouettes than Shape of Motion [68] and MoSca [29].

**Limitations.** Our dynamic representation struggles with objects that move predominantly outside the viewing frustum and later reappear, due to insufficient photometric constraints for Motion Scaffolds. This causes optimization to be dominated by regularization terms, often leading to undesired deformations. This issue is an inherent limitation also observed in MoSca [29]. Additionally, similar to most monocular dynamic view synthesis methods, our system requires per-scene test-time optimization, thereby limiting its applicability in real-time scenarios. Furthermore, ProDyG cannot effectively handle large changes in novel viewpoints, which would necessitate generative models or data-driven priors to hallucinate unseen regions. These challenges highlight key directions for future work.

| | Shape of Motion [68] | DynOMo [56] | MoSca [29] | Gaussian Marbles [61] | ProDyG (Ours) | ProDyG (Ours) | ProDyG (Ours) | ProDyG (Ours) |
|---|---|---|---|---|---|---|---|---|
| Online Reconstr. | ✗ | ✓ | ✗ | ✗ | ✓ | ✓ | ✓ | ✓ |
| Online Tracking | ✗ | ✗ | ✗ | ✗ | ✗ | ✓ | ✗ | ✓ |
| RGB-only | ✗ | ✗ | ✗ | ✗ | ✗ | ✗ | ✓ | ✓ |
| PSNR↑ | 17.43 | 11.98 | **18.44** | 16.00 | 17.65 | 17.87 | 15.41 | 15.40 |
| SSIM↑ | 0.591 | 0.436 | **0.666** | - | 0.634 | 0.643 | 0.603 | 0.582 |
| LPIPS↓ | **0.303** | 0.748 | 0.311 | 0.437 | 0.390 | 0.377 | 0.462 | 0.492 |

Table 2: **Novel View Synthesis Evaluation on iPhone Dataset [12]**. All results are averaged over the 5 sequences evaluated in [68], with the standard 2x downsampling. All methods except [68] are evaluated without ground truth camera poses. Best results are highlighted as first , second , third . Our method shows superior PSNR and SSIM over the offline Shape of Motion and falls short of the state-of-the-art offline method MoSca [29] by a small margin under the extra constraints of online reconstruction and tracking. When tested with precomputed camera poses, ProDyG outperforms the only online competitor DynOMo [56] by a significant advantage. Notably, our method still works reasonably well with RGB-only input while being online.

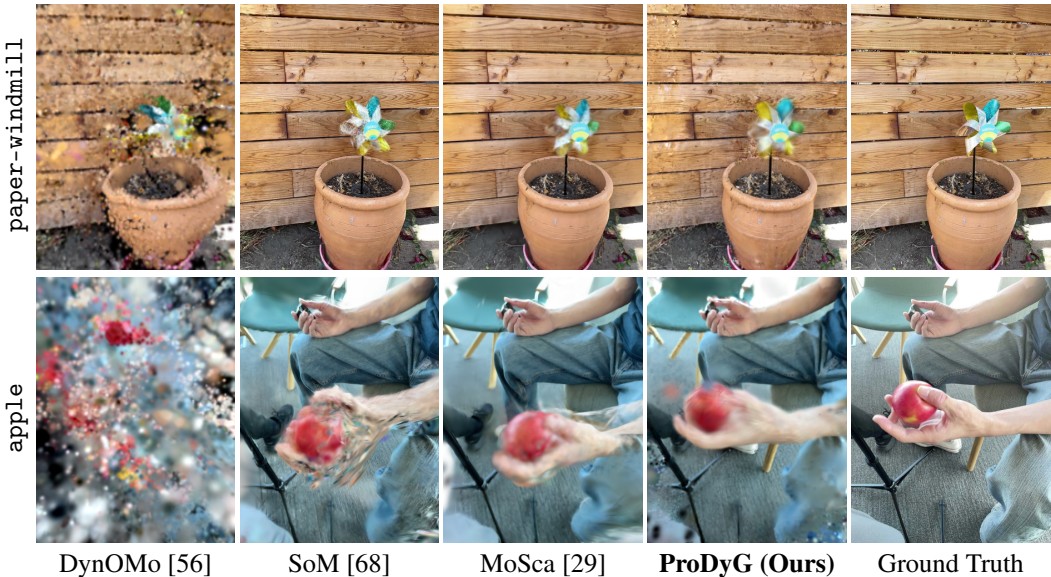

DynOMo [56]     SoM [68]     MoSca [29]     **ProDyG (Ours)**     Ground Truth

Figure 4: **Qualitative Novel View Synthesis Results on iPhone [12].** All methods are trained with access to ground truth depth maps. Dynamic objects reconstructed by our method tend to show more accurate silhouettes than [68] and [29]. Note that ProDyG is the only one among the methods to perform both tracking and reconstruction online.

# 5   Conclusion

We proposed ProDyG, a progressive dynamic 3D reconstruction framework that meets four key criteria for practical deployment: online operation, global pose and map consistency, detailed appearance and geometry modeling through 3D Gaussian Splatting, and flexibility to operate with either RGB or RGB-D input. Our novel flow-based motion-mask prediction integrated with the SLAM backend enables robust camera tracking in dynamic environments, while our online dynamic reconstruction pipeline updates and optimizes Motion Scaffolds and dynamic Gaussians in a progressive manner. Our experiments demonstrate that ProDyG achieves competitive performance in both tracking and novel view synthesis.

# Acknowledgments

We thank the Computer Vision Lab at ETH Zurich for providing the GPU compute resources used in this research, with approval from Ender Konukoglu. Erik Sandström conducted this work as part of his employment at Google Zurich, which supported his participation in this research; Google had no direct role in the project's design, execution, or analysis. Shi Chen acknowledges support from the NeurIPS 2025 Scholar Award and thanks Stefan Leutenegger for additional travel support.

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
