# OpenReview forum: "ProDyG: Progressive Dynamic Scene Reconstruction via Gaussian Splatting from Monocular Videos"
_NeurIPS.cc/2025/Conference — NeurIPS 2025 poster_

### Official Review · Reviewer_KJPL · 2025-06-14

**Clarity:** 4
**Significance:** 3
**Originality:** 2
**Rating:** 5
**Confidence:** 4

**Summary:**

The submission tackles online monocular dense dynamic scene reconstruction. The key differentiator to prior work is that depth input is optional. Similar to Motion Scaffolds, the method uses a number of pretrained models to extract depth, segmentation masks, and optical flow. It combines all this information with a motion scaffold to obtain a dynamic reconstruction. Unlike Motion Scaffolds, the proposed method works in an online setting. It borrows ideas like its keyframe handling and static reconstruction from Splat-SLAM to estimate camera poses online, which originally only handles static scenes.

**Questions:**

See the major weaknesses above since they are intertwined. All major weaknesses are experimental. Although I enjoyed this submission, I'd like to see video results, ablations, and better tracking evaluations to recommend acceptance. Since the rebuttal format does not allow for videos, I am not sure that my concerns can be addressed in a rebuttal.

**Ethical Concerns:**

["NO or VERY MINOR ethics concerns only"]

**Final Justification:**

The rebuttal addresses all of my concerns well enough, given the limitations of the rebuttal setting. I read the other discussions as well and I didn't find anything objectionable. I trust the authors to follow up on their promises in a revised version. I'll raise my rating to accept.

**Limitations:**

The method is global for the static part of the scene but the dynamic part is only global "accidentally" by stitching together local dynamics, i.e. dynamic loop closure / track re-identification after prolonged invisibility is not tackled beyond accidental re-identification due to simple regularizers like ARAP. Please make this clearer in the Limitations.

**Quality:**

3

**Strengths And Weaknesses:**

Strengths:

The problem setting is very interesting and challenging. It is also timely, with two concurrent works [8,18], see Table 1 in the supplement and Table 1 in the main paper.

The Related Work section is very thorough and to the point.

The Method section is written well.

The method is designed quite well, and extends prior work in a sensible manner.

The authors promise a code release.



Weaknesses:

The evaluation has very few qualitative results. There are no video comparisons to other methods.

Qualitative video results for Table 2 would be good to get a sense of how important depth input is and how well the method estimates poses when they are not given.

Other than Table 2, there appear to be no ablations despite the rather involved design of the method. For example, does the motion mask algorithm need to be this complicated since it ultimately only seeds queries to SAM2 (and removes dynamic components for pose estimation but being over-eager there doesn't seem like a problem)?

Something seems off about the qualitative comparisons in Fig. 4: Why are the results of DynOMo so bad? The results on their project page look much better, including on the same scene.

Presumably the tracking performance is evaluated on static and dynamic components? If so, the static components would heavily dominate the results, as seen by WildGS-SLAM (which apparently is only robust against dynamics and does not reconstruct them) dominating in Table 1. Please provide masked results where only the dynamic points are evaluated (even using coarse motion masks should give enough signal but preferably use the fine motion masks).

*** Minor ***

How are poses in between keyframes estimated? Presumably, the method runs until it hits the next keyframe and then goes back and processes all non-keyframes in between in an online fashion? This could be clearer in the text.

Why is the method so slow, at 10-30 sec per frame? How is this split among the components? Presumably the pretrained models make up the bulk of that because I don't see how the presented method takes that much time otherwise?

Why is static Splat-SLAM treated as the main tracking baseline? It obviously does not handle dynamics.

The trajectories are hard to see in the video results.

---

> ### Author Rebuttal · Authors · 2025-07-30
>
> Thank you for the input! We respond to the weaknesses, questions and limitations you raised as follows:
>
> ## Weaknesses
>
> > “The evaluation has very few qualitative results. There are no video comparisons to other methods.”
>
> **Due to the volume limit of only 9 one-columned pages**, we were only able to show a few qualitative results in the paper. **We have many more which we plan to release on our upcoming project page**, including predicted camera trajectories, training-view rerenderings, novel-view renderings on the iPhone dataset (all sequences including those not providing GT test views, e.g. handwavy, haru-sit…), the Bonn Dynamic dataset and the TUM RGB-D Dynamic dataset. And thank you for pointing out the lack of video comparisons. **We will also put video comparisons** with some important baseline methods (offline: Shape of Motion, MoSca, etc., online: DynOMo, DynaMoN, etc.) **on our project page**.
>
> > “Qualitative video results for Table 2 would be good to get a sense of how important depth input is and how well the method estimates poses…”
>
> Thank you for the suggestion. **We will also show these video comparisons on our project page** to give readers a better idea of the influence of depth and pose inputs.
>
> > “Other than Table 2, there appear to be no ablations… does the motion mask algorithm need to be this complicated... being over-eager there doesn't seem like a problem)?”
>
> Thank you for pointing this out! Yes, for tracking/pose estimation it is okay for the motion mask algorithm to be over-eager to some extent as long as it leaves enough static regions for the DBA to function well. Of course, if the algorithm misses some dynamic objects in the scenes, the tracking performance degrades. The comparison of ProDyG and Splat-SLAM in Table 1 supports this statement. **We believe the importance of having accurate fine motion masks lies more in the reconstruction/NVS side.** To demonstrate the importance of our algorithm design for SAM2 prompting, **we conducted an ablation study focused on NVS quality**: We compared the NVS performance on the paper-windmill sequence from the iPhone dataset under three different setups:
> **1.** ProDyG (using our residual-flow-based prompting)
> **2.** ProDyG, but replacing the residual-flow-based prompting with a similar approach based on optical flow applied in DynaGSLAM [34]. Specifically, we compute and threshold the magnitude gradient of optical flow estimated by the SLAM backend to detect object edges, and close the shapes using dilations and erosions. After that, we prompt SAM2 at the centroids of connected regions in the resulting binary masks. We will include more technical details of this algorithm in the camera ready version of our supplementary material.
> **3.** Splat-SLAM (no motion masking at all)
> We show the results in the table below:
>
> | Setup | PSNR  | SSIM | LPIPS |
> | :--: | :--: | :--: | :--: |
> | **1 (Ours)** | **20.05** | **0.62** | **0.26** |
> | 2 | 19.18 | 0.51 | 0.31 |
> | 3 | 17.49 | 0.54 | 0.31 |
>
> **Our SAM2 prompting algorithm clearly outperforms the substitute method and the case where no motion masking is applied.** The substitute method (Setup 2) tends to over-segment, mis-identifying some static foreground objects (e.g. the flower pot) as dynamic ones. As a result, the static objects are reconstructed with some faulty motion, causing poor NVS quality in those regions. In comparison, **our residual-flow-based method manages to decouple true scene motion from camera-induced motion and thus more accurate motion masks, resulting in superior NVS results.**
>
> > “Something seems off about the qualitative comparisons in Fig. 4: Why are the results of DynOMo so bad? …”
>
> Thank you for the question. We need to point out that **the renderings on DynOMo’s project page are all from training views, while what we show in Fig. 4 are novel-view renderings**. According to our observations running DynOMo, their NVS is constantly poor, especially later in sequences (will be shown in the video comparisons on our project page). They don’t seem to employ any effective constraints on the Gaussian primitives along the ray directions to ensure 3D consistency. After all, DynOMo only guarantees good 2D tracking.
>
> > “Presumably the tracking performance is evaluated on static and dynamic components? ...”
>
> We believe **there has been a misunderstanding between point tracking**, which measures how  dynamic objects move, **and camera tracking**, which estimates the camera extrinsics. **What we show in Table 1 are the camera tracking results**, which we measure using ATE RMSE. We will add point tracking results evaluated on the iPhone dataset (only on dynamic points) in the camera ready version. However, we would like to stress that even without point tracking evaluation, our NVS results shown in Table 2, Figure 4 and the supplementary video are already able to demonstrate how well ProDyG models scene dynamics to a great extent. Moreover, NVS is usually a more difficult task than 2D point tracking due to the need to maintain 3D consistency. DynOMo, as we show in Table 2 and Figure 4, is a typical example of performing very well in 2D point tracking while being extremely poor at NVS. This is why we prioritized NVS evaluation over point tracking given the limited space of the main paper.
>
> ### Minor
>
> > “How are poses in between keyframes estimated? ...”
>
> Apologies for not explaining this clearly enough. Your interpretation is absolutely correct.
>
> > “Why is the method so slow… How is this split among the components? ...”
>
> **The geometry optimization of the Motion Scaffolds, the photometric optimization of dynamic Gaussians, and CoTracker tracking are the three main contributors of the runtime.** We show in Table S2 that we can cut down a major part of the runtime by reducing the number of iterations for the geometry and photometric optimization and number of initial tracks for CoTracker, with minimal performance loss. Here, we also show a clearer **component-wise runtime breakdown** when running ProDyG on the paper-windmill sequence in the iPhone dataset below (Here, we split the total runtime into time for tracking / mapping, and the time for mapping is further split into time for the geometry optimization, the photometric optimization, the CoTracker3 runtime, and time for the rest of the operations, e.g. data loading.).
>
> | Component                                        | Time (s) |
> | :----------------------------------------------- | -------: |
> | **Total**                                       | 8052     |
> | ├─ Tracking                                      | 22       |
> | └─ Mapping                                       | 8030     |
> | &nbsp;&nbsp;&nbsp;&nbsp;&nbsp;&nbsp;├─ Geometry optimization         | 1230     |
> | &nbsp;&nbsp;&nbsp;&nbsp;&nbsp;&nbsp;├─ Photometric optimization      | 5424     |
> | &nbsp;&nbsp;&nbsp;&nbsp;&nbsp;&nbsp;├─ CoTracker3 running time      | 785      |
> | &nbsp;&nbsp;&nbsp;&nbsp;&nbsp;&nbsp;└─ Rest                                      | 591      |
>
> > “Why is static Splat-SLAM treated as the main tracking baseline? ...”
>
> Thank you for pointing this out! Indeed, “main baseline” is confusing. This is only meant to show that our modification based on Splat-SLAM manages to make it handle dynamics. We will improve the wording here in the camera ready version.
>
> > “The trajectories are hard to see in the video results.”
>
> Apologies. Currently both the camera and Gaussian trajectories are visualized by inserting extra Gaussians and rendering them along with other Gaussians. We will improve the visualization code before putting the videos on our project page.
>
> ## Questions
>
> > “See the major weaknesses above since they are intertwined…”
>
> Thank you very much for the insightful review! We tried our best within the short rebuttal period to respond to your questions, and to address some of the issues you raised, we added **an ablation of the motion mask generation algorithm** to demonstrate its importance to reconstruction/NVS. We also provide **an extra component-wise runtime breakdown**. Unfortunately, **we are not allowed to show more qualitative results in the rebuttal, but we will instead put them on our upcoming project page**. We hope our response and additional experimental results address your concerns more or less.
>
> ## Limitations
>
> > “The method is global for the static part of the scene but the dynamic part is only global "accidentally" by stitching together local dynamics...”
>
> Agreed. ProDyG is yet to handle re-identification of dynamic objects after long invisibility, and this is indeed an interesting and important direction for future work. We will clarify this in the camera ready version.
>
> ## References
>
> [34] Li, R.B., Shaghaghi, M., Suzuki, K., Liu, X., Moparthi, V., Du, B., Curtis, W., Renschler, M., Lee, K.M.B., Atanasov, N., et al.: Dynagslam: Real-time gaussian-splatting slam for online rendering, tracking, motion predictions of moving objects in dynamic scenes. arXiv preprint arXiv:2503.11979 (2025)

---

> > ### Comment · Reviewer_KJPL · 2025-08-05
> > **Involved Method**
> >
> > Thank you for the rebuttal, it addresses all of my points well except for the point that reviewer KrLA made a bit more clearly than me: apart from speed considerations, the rebuttal seems to imply that there is "some functional overlap in terms of finding spatial correspondences, so in theory there is room for simplification." Can you please provide a discussion about the advantages of the more involved design? I'm not sure what exactly I'm trying to get at, unfortunately, so it is possible that this is somehow the best design but as the rebuttal mentions, it seems like there is some unnecessary redundancy there.

---

> > > ### Author Response · Authors · 2025-08-06
> > > **Justification for Method Design**
> > >
> > > Thank you for the feedback! We would like to emphasize that **our seemingly "redundant" method design is not unnecessary, but rather a locally optimal one**. This is because **DROID-SLAM, SAM2 and CoTracker3, though all somehow finding correspondences, are optimized for different tasks and are thus not interchangeable**.
> > >
> > > DROID-SLAM estimates **low-resolution** optical flow **between keyframes** at very **low computational cost**, and has been proved **more robust and accurate in tracking than more recent SLAM systems based on denser correspondences** (e.g. NICER-SLAM [93]). Therefore, the correspondences used here do not need to be as dense as those in SAM2 or CoTracker3. In ProDyG, the low-resolution optical flow is also used for motion mask generation. We have shown through the additional ablation study that our prompting strategy based on residual flow outperforms the DynaGSLAM one. Could we potentially replace this with CoTracker3? Yes, but it would be suboptimal due to its **noisiness** and **extra computational overhead when having to track all points** instead of only dynamic ones, as mentioned in response to reviewer KrLA.
> > >
> > > Meanwhile, **our dynamic reconstruction method requires per-frame fine-grained motion masks** to identify dynamic regions, **and dense (spatially and temporally) 2D tracks** to initialize motion scaffolds and supervise dynamic Gaussians. These requirements cannot be met by the low-resolution, inter-keyframe optical flow from DROID-SLAM, making it unsuitable as a replacement for SAM2 or CoTracker3.
> > >
> > > Although SAM2 and CoTracker3 have more in common in that they both densely track similar points over time in semantics or appearances, they are both very well tuned for their respective tasks (i.e. segmentation and point tracking) through large-scale training, and we leverage this by loading their pretrained models. While a more elegant, unified architecture that shares features across tasks (e.g. motion mask generation and point tracking) could theoretically be designed, it would be **unrealistic to expect it to match the performance of SAM2 and CoTracker3 in both tasks without extensive training**. And even if we manage to do that, **the runtime gain would be marginal**, since the majority of time is spent on geometry and photometric optimization.
> > >
> > > We believe the above reasons justfies our current method design.
> > >
> > > References:
> > >
> > > [93] Zhu, Z., Peng, S., Larsson, V., Cui, Z., Oswald, M.R., Geiger, A., Pollefeys, M.: NICER-SLAM: Neural Implicit Scene Encoding for RGB SLAM. In: International Conference on 3D Vision (3DV) (March 2024)

---

> > > > ### Comment · Reviewer_KJPL · 2025-08-06
> > > > **Thank you**
> > > >
> > > > Thank you. If possible, please add this in some manner to the supplementary PDF since this kind of reasoning is quite helpful for others in the field. I'll raise my score to accept.

---

> > > > > ### Author Response · Authors · 2025-08-06
> > > > > **Thank You**
> > > > >
> > > > > Thank you very much for the insightful review which makes our work more complete! We will definitely add the justification to the camera-ready supplementary as you suggested.

---

### Official Review · Reviewer_KrLA · 2025-07-02

**Clarity:** 3
**Significance:** 3
**Originality:** 3
**Rating:** 3
**Confidence:** 5

**Summary:**

This paper presents a complete online system that simultaneously tracks the camera and reconstructs both static backgrounds and dynamic foregrounds using 3D Gaussian Splatting (3DGS). The proposed pipeline leverages motion scaffolds to propagate 3DGS representations in real time and handles dynamic environments by integrating coarse and fine-grained motion cues.

**Questions:**

It would be helpful if the authors could make the method section clearer and more organized.
It would also be great to include results on long video sequences.

**Ethical Concerns:**

["NO or VERY MINOR ethics concerns only"]

**Final Justification:**

After reviewing the other reviewers’ feedback, I believe this paper is on the borderline between acceptance and rejection, and I am comfortable with either outcome.

**Limitations:**

Yes.

**Paper Formatting Concerns:**

No, I don't have concern.

**Quality:**

3

**Strengths And Weaknesses:**

Strengths:

Effective Dynamic Masks:
The system combines DROID-SLAM for coarse dynamic masking and SAM2 for fine-grained motion segmentation. This combination is both practical and efficient, enabling robust foreground-background separation without requiring heavy supervision or extensive engineering.

Practicality and Real-World Application:
While the system is somewhat high-level and system-oriented, it targets real-world challenges in dynamic scene reconstruction. The integration of robust tracking and dynamic object modeling makes it highly applicable for robotics and AR/VR use cases. The effort to build a usable system in this underexplored direction is commendable.


Weaknesses and Questions:

1. Unclear Initialization Strategy:
The initialization phase of the system is not clearly described. It would be helpful to elaborate on how camera poses are recovered at the start and how the initial 3D reconstruction is built.
2. Pose Estimation in Dynamic Scenes:
It's not fully explained how the system estimates accurate poses in the presence of dynamic elements. What is the robustness mechanism during scenes with large dynamic foreground motion?
3. Optical Flow Redundancy:
The system uses DROID-SLAM for motion masking and CoTracker-3 for generating long-term dense 2D trajectories. A natural question arises: why not rely solely on CoTracker to distinguish dynamic and static regions? It could simplify the pipeline and reduce dependencies.
4. System Redundancy and Efficiency:
The current design involves multiple overlapping components—DROID-SLAM for masks, SAM2 for refinement, and CoTracker for 2D tracking. While each tool contributes to robustness, the pipeline appears somewhat redundant. A discussion on potential simplification or trade-offs in performance vs. efficiency would strengthen the work.
5. Real-Time Performance and Memory Profiling:
The paper claims real-time capability, but there is no detailed evaluation of runtime or memory usage across modules. A table or analysis comparing the runtime breakdown and GPU memory requirements of each stage would make the real-time claim more convincing.
6. Pose and Motion Optimization Details:
The pose optimization process is not well explained. How does the system jointly recover camera poses and dynamic object trajectories? Is the optimization based on sparse or dense correspondences, and is it integrated into the 3DGS update?

---

> ### Author Rebuttal · Authors · 2025-07-30
>
> Thank you for your insightful input! Since your questions mainly concern specific details of the method section and the motivation behind our design choices, we believe these can be effectively addressed by our responses below (including a module-wise runtime breakdown):
>
> ## Weaknesses and Questions
>
> > **1.** “Unclear Initialization Strategy”
>
> Sorry for not explaining clearly enough how the initialization phase works. **We chose to focus on our contributions which pertains to updating the camera poses and the scene representation progressively.** Roughly speaking, the way we initialize the camera poses and static Gaussians is similar to that of Splat-SLAM [54]. Specifically, the system first builds a factor graph by adding keyframes (a new keyframe is added when the average optical flow magnitude exceeds a preset threshold compared to the last keyframe), and as soon as the number of keyframes reaches a certain number (set to 12 in all our experiments), the system bootstraps the collected keyframes to get their camera poses. After the bootstrapping, the static Gaussians are initialized and optimized using the latest keyframe (RGB image and depth map estimated by the SLAM backend). As for the initialization of dynamic reconstruction, we do it similarly to MoSca [29] using all frames (RGB images, estimated camera poses, depth maps and motion masks) up to the last keyframe used for initial bootstrapping. Specifically, we run CoTracker3 [20] within the identified dynamic regions to get dense 2D tracks and sample a subset of them to lift into 3D and initialize motion scaffolds. Then, geometry and photometric optimizations are carried out to get the initial dynamic reconstruction. We will clarify this part further in the camera ready version.
>
> > **2.** “Pose Estimation in Dynamic Scenes”
>
> Generally, ProDyG is robust against dynamic foreground motion by predicting motion masks and suppressing the confidence weight of identified dynamic regions to zero in subsequent DBA iterations. After each DBA iteration, the system computes (see Eqn. 2) and thresholds the “residual flow” at the top 20% to get a coarse motion mask. Then, the coarse motion mask filters out potentially dynamic regions in the next DBA iteration for more robust egomotion estimation. As the DBA iterations go on, the coarse motion masks are gradually updated, and the pose estimation becomes increasingly robust against foreground dynamics because of the improvement of the masks. When the mapping module is triggered, we further refine the coarse motion masks by prompting SAM2 [50] (see “Semantic-Guided Motion Mask Refinement” in 3.1 and in Supplementary Material A), leveraging the power of semantic segmentation. Afterwards, we replace the coarse masks using the refined ones for more accurate, confidence weighted suppression, in subsequent iterations of DBA, which includes local BA, global BA and loop BA (please refer to Splat-SLAM [54] regarding these different types of Bundle Adjustment).
>
> > **3.** “Optical Flow Redundancy”
>
> Our observation is that the internal flow representation of DROID-SLAM [66] is rather low-resolution, but more robust for pose estimation. We use CoTracker3 [20] to predict dense 2D tracks which we use to initialize motion scaffolds, and to provide extra supervision for dynamic Gaussians. We choose not to use CoTracker3 also for dynamic-static separation for mainly two reasons: a) **CoTracker point tracking results can be noisy**, especially in textureless regions, which can lead to suboptimal motion masking (the low-resolution optical flow estimated by DROID-SLAM is more robust in this sense); b) Currently, we only use Cotracker to track points within the motion masks. **If we tracked all points densely, it would add a considerable computation overhead.**
>
> > **4.** “System Redundancy and Efficiency”
>
> Thank you for the insightful question! Indeed, the optical flow estimation in DROID-SLAM [66], SAM2 [50] for semantic segmentation of dynamic objects and CoTracker3 [20] for dense 2D point tracking have some functional overlap in terms of finding spatial correspondences, so in theory there is room for simplification. In practice, however, considering the main runtime bottleneck is geometry and photometric optimization of Motion Scaffolds and dynamic Gaussians (see the runtime breakdown we provide in response to 5), **the efficiency gain of removing this redundancy would be marginal.**
>
> > **5.** “Real-Time Performance and Memory Profiling”
>
> We need to first clarify that **we don’t claim real-time capability**. In addition to Table S2 where we report the overall runtime and GPU memory consumption, here we provide an extra **module-wise runtime breakdown** when running ProDyG on the paper-windmill sequence in the iPhone dataset (we split the total runtime into time for tracking / mapping, and the time for mapping is further split into time for the geometry optimization, the photometric optimization, the CoTracker3 [20] runtime, and time for the rest of the operations such as data loading):
>
> | Component                                        | Time (s) |
> | :----------------------------------------------- | -------: |
> | **Total**                                       | 8052     |
> | ├─ Tracking                                      | 22       |
> | └─ Mapping                                       | 8030     |
> | &nbsp;&nbsp;&nbsp;&nbsp;&nbsp;&nbsp;├─ Geometry optimization         | 1230     |
> | &nbsp;&nbsp;&nbsp;&nbsp;&nbsp;&nbsp;├─ Photometric optimization      | 5424     |
> | &nbsp;&nbsp;&nbsp;&nbsp;&nbsp;&nbsp;├─ CoTracker3 running time      | 785      |
> | &nbsp;&nbsp;&nbsp;&nbsp;&nbsp;&nbsp;└─ Rest                                      | 591      |
>
> > **6.** “Pose and Motion Optimization Details”
>
> In the ProDyG pipeline, camera pose estimation and dynamic reconstruction are separated. At a high level, camera poses are estimated as follows: low-resolution optical flow is computed between keyframes and keyframe poses and depth maps are optimized to explain the flow in static regions indicated in the estimated motion masks (see our response to 2) as much as possible. The dynamic object trajectories are mainly recovered from the CoTracker3-estimated [20] dense tracks and photometric clues. Therefore, both pose estimation and dynamic reconstruction are based on dense correspondences, but the optical flow used for pose estimation is at a lower resolution, and yes, it is integrated into the 3DGS update since we use a photometric loss and a 2D track loss during the photometric optimization phase.
>
> > “It would be helpful if the authors could make the method section clearer and more organized. It would also be great to include results on long video sequences.”
>
> Thank you for the advice. We will improve the structure of the method section to make it clearer in the camera ready version. In terms of video lengths, the longest video among those we conduct experiments on is “freiburg3_walking_halfsphere” in the TUM RGB-D dataset, which is 35.81 seconds long. We believe this is a sufficiently long sequence to show the capabilities of our method.
>
> ## References
>
> [20] Karaev, N., Makarov, I., Wang, J., Neverova, N., Vedaldi, A., Rupprecht, C.: Cotracker3: Simpler and better point tracking by pseudo-labelling real videos. arXiv preprint arXiv:2410.11831 (2024)
> [29] Lei, J., Weng, Y., Harley, A., Guibas, L., Daniilidis, K.: Mosca: Dynamic gaussian fusion from casual videos via 4d motion scaffolds. arXiv preprint arXiv:2405.17421 (2024)
> [50] Ravi, N., Gabeur, V., Hu, Y.T., Hu, R., Ryali, C., Ma, T., Khedr, H., Rädle, R., Rolland, C., Gustafson, L., et al.: Sam 2: Segment anything in images and videos. arXiv preprint arXiv:2408.00714 (2024)
> [54] Sandström, E., Tateno, K., Oechsle, M., Niemeyer, M., Van Gool, L., Oswald, M.R., Tombari, F.: Splat-slam: Globally optimized rgb-only slam with 3d gaussians. arXiv preprint arXiv:2405.16544 (2024)
> [66] Teed, Z., Deng, J.: Droid-slam: Deep visual slam for monocular, stereo, and rgb-d cameras. Advances in neural information processing systems 34, 16558–16569 (2021)

---

> > ### Comment · Area_Chair_hVXW · 2025-08-08
> >
> > Dear Reviewer,
> > Please consider the reply by the authors/other reviews and see whether or not this would have any influence on your original rating. Please also note that you must acknowledge having read the reply and enter your final rating and justification before the deadline. Respectfully, your AC

---

> ### Comment · Reviewer_KrLA · 2025-08-08
>
> Thank you for your thoughtful and comprehensive responses. I appreciate the clear breakdowns and the additional details provided across all components. I’ve reviewed your clarifications carefully, and I offer the following comments:
>
> 1. Unclear Initialization Strategy:
> Your explanation clarifies the bootstrapping process for both static and dynamic scene elements. The reference to Splat-SLAM and MoSca, along with your use of CoTracker3 for initializing motion scaffolds, provides a coherent picture of your design. Clarifying this flow in the camera-ready version will significantly improve readability.
>
> 2. Pose Estimation in Dynamic Scenes:
> The iterative refinement strategy using residual flow and semantic-guided masks is a strong design choice. Integrating SAM2 to refine masks post-DBA and improve pose robustness is well justified. The layered approach is now much clearer and demonstrates robustness in dynamic settings.
>
> 3. Optical Flow Redundancy:
> Thank you for detailing your reasoning behind combining low-res DROID-SLAM flow and CoTracker3 for different roles. The trade-off between robustness and density, and the choice to limit CoTracker3 usage to motion regions, is a reasonable and pragmatic design.
>
> 4. System Redundancy and Efficiency:
> I appreciate the honest acknowledgment of the potential redundancy in using multiple flow and segmentation modules. Your justification, grounded in the actual runtime bottlenecks (geometry and photometric optimization), makes sense. A brief discussion of this trade-off in the paper would help clarify your design choices to readers.
>
> 5. Real-Time Performance and Memory Profiling:
> Thanks for providing the module-wise runtime breakdown—it is very helpful. While ProDyG is not real-time, the detailed analysis shows a clear understanding of where computation is spent. This will benefit others attempting to build upon or optimize your system.
>
> 6. Pose and Motion Optimization Details:
> The separation of camera pose estimation and dynamic reconstruction is now well explained. Using dense correspondences for both, but different flow resolutions and roles, is a thoughtful design. The explanation of how optical flow integrates with 3DGS updates through photometric loss provides sufficient clarity.
>
> General Suggestions:
> Method Section: I strongly support your plan to restructure and clarify the method section. Making initialization, dynamic modeling, and optimization phases more modular in description will enhance clarity.
>
> Longer Videos: While your current experiments include sequences up to ~36 seconds, extending evaluation to longer or more complex sequences in future work could better showcase temporal consistency and scalability.
>
> Tables Results: It will be great to explain the potential reason why our method didn't get the best performance to beat the SOTA methods (It looks most of trajectory is the second best). You don't need to make you method best but provide necessary explanation will be great for reader to improve our system.
>
> Online and Real-time Discussion: sometimes, I get confused by the difference between online and real-time's definition. In the SLAM community, most researchers assume online system is a real-time system. It might causes some confusion for different community to review our paper. ^ ^
>
> Overall assessment: I think the author address my concerns and very appreciate the author's effort on rebuttal! Thank you!

---

> > ### Author Response · Authors · 2025-08-08
> > **Thank You**
> >
> > Thank you for your feedback! We are glad that our rebuttal has addressed all your concerns.

---

### Official Review · Reviewer_JEry · 2025-07-03

**Clarity:** 4
**Significance:** 3
**Originality:** 3
**Rating:** 5
**Confidence:** 3

**Summary:**

This is a new online SLAM system that performs dense reconstruction of both static and dynamic scene elements, using gaussian splats. The emphasis is not so much on novel components as combining them in an intelligent and novel way into an end-to-end system with good performance.

**Questions:**

Must field-of-view / intrinsics be supplied or does ProDyG optimize them (sorry if I missed this in the paper)? A column for this could be a nice addition to Table S1 ("Method Classification").

l.101,113 describe threshold motion masks at 20%. Does this tell us something about how much dynamic content can be present before the method stops working successfully?

The MegaSaM system is claimed to also work well on scenes with low parallax (i.e. very little camera motion or motion with primarily rotation) that traditional SLAM systems tend to fail on. How does ProDyG fare on such cases?

**Ethical Concerns:**

["NO or VERY MINOR ethics concerns only"]

**Final Justification:**

My questions were answered very satisfactorily. After reading the response and the other authors' reviews, I retain my rating of Accept.

**Limitations:**

Yes, but it would also be informative (in the supplementary video) to see some qualitative examples of failure modes.

**Paper Formatting Concerns:**

I didn't exactly understand the claim at line 38 ("first online RGB-only method"). Table S1 includes several methods which are classified as both online and RGB-only. Perhaps some other condition could be added to make this claim more specific.

There are lots of capitalization errors in the References ("3d", "slam", etc).

**Quality:**

3

**Strengths And Weaknesses:**

[Strengths] The writing is excellent throughout, with clear and concise descriptions and a minimum of tedious puffery. This is a complex system with many moving parts and yet the paper manages both to describe it at a high level and to provide many specific details (with many more in the supplementary material). Reimplementing such a system is a major task but with the promise to release code, it should be possible to use it and base future research on it.

[Strength] Both the overall approach and the many individual design decisions are well-motivated. To call out a couple in particular: I like the method of using flow to generate coarse masks and a segmentation model to refine them, and I was impressed to see that tracking and motion scaffolds have been added without breaking the online nature of the system.

[Mixed] Evaluation is performed against a wide selection of baselines, and covers both tracking and novel view synthesis. However, the choice of datasets is somewhat limited, as are the metrics (no rotation error metrics are reported). Runtime numbers are also given (in the supplementary materials), though here there is not much comparison to other methods. (I was also curious to know whether the runtime might be competitive with other SLAM systems when used in a tracking-only manner.)

[Weakness] There are not many qualitative examples, and no qualitative comparison of tracks. Are the top methods all similar enough that there is no interesting qualitative difference?

---

> ### Author Rebuttal · Authors · 2025-07-30
>
> Thank you for your input! We respond as follows:
>
> ## Strength and Weaknesses
>
> > “[Mixed] Evaluation is performed against a wide selection of baselines... the choice of datasets is somewhat limited, as are the metrics (no rotation error metrics are reported)... whether the runtime might be competitive with other SLAM systems when used in a tracking-only manner.”
>
> Possible options of datasets are limited, when requiring dynamic objects and known camera motion. Regarding the metrics for tracking, we could have reported rotation error for our method, but as this is not reported in the vast majority of baseline methods we compare our method to, we ended up only reporting ATE RMSE.
> When used in a tracking-only manner, our runtime is almost the same as Splat-SLAM [54] (tracking-only, ~10 FPS). To make this clearer, we show below a **component-wise runtime breakdown** when running ProDyG on the paper-windmill sequence in the iPhone dataset:
>
> | Component                                        | Time (s) |
> | :----------------------------------------------- | -------: |
> | **Total**                                       | 8052     |
> | ├─ Tracking                                      | 22       |
> | └─ Mapping                                       | 8030     |
> | &nbsp;&nbsp;&nbsp;&nbsp;&nbsp;&nbsp;├─ Geometry optimization         | 1230     |
> | &nbsp;&nbsp;&nbsp;&nbsp;&nbsp;&nbsp;├─ Photometric optimization      | 5424     |
> | &nbsp;&nbsp;&nbsp;&nbsp;&nbsp;&nbsp;├─ CoTracker3 running time      | 785      |
> | &nbsp;&nbsp;&nbsp;&nbsp;&nbsp;&nbsp;└─ Rest                                      | 591      |
>
> Here, we split the total running time into time for tracking / mapping, and the time for mapping is further split into time for the geometry optimization, the photometric optimization, the CoTracker3 [20] running time, and time for the rest of the operations (such as data loading).
>
> > “[Weakness] There are not many qualitative examples, and no qualitative comparison of tracks. Are the top methods all similar enough that there is no interesting qualitative difference?”
>
> Yes. For example, our predicted camera trajectories tend to look very similar to what WildGS-SLAM predicts (and also to GT trajectories) when plotted in 2D. Therefore, we prioritized showing qualitative results of NVS given the limited space. We generated additional qualitative videos which will be published on our project website.
>
> ## Questions
>
> > “Must field-of-view / intrinsics be supplied…”
>
> Under the current design, the intrinsics need to be supplied beforehand. However, since MoSca, an important component of our method, supports the joint optimization of camera intrinsics, it should be possible to extend ProDyG to handle it.
> Thank you also for your suggestion to add a column about this in Table S1! We will do so in the camera ready version.
>
> > “l.101,113 describe threshold motion masks at 20%. Does this tell us something about how much dynamic content can be present...”
>
> Yes, this is a heuristic hyperparameter setting and it does provide a prior about the general proportion of dynamic content present in the frames. However, please note that this is a global hyperparameter applied to all sequences we report experimental results on (i.e. we did not tune it sequence-by-sequence).
>
> > “The MegaSaM system is claimed to also work well on scenes with low parallax… How does ProDyG fare on such cases?”
>
> Thank you for bringing this up! Our observation is that ProDyG’s tracking performance can degrade in such cases, as triangulation becomes degenerate and we do not employ any special designs to address this issue as MegaSaM does. That said, it should be easy to address this issue in a similar fashion since both ProDyG and MegaSaM are based on DROID-SLAM.
>
> ## Limitations
>
> > “Yes, but it would also be informative (in the supplementary video) to see some qualitative examples of failure modes.”
>
> Thank you for the suggestion! Unfortunately, we are not allowed to show them here in the rebuttal. We will instead show some qualitative examples of failure modes on our upcoming project page.
>
> ## Paper Formatting Concerns
>
> > “I didn't exactly understand the claim at line 38 ('first online RGB-only method')…”
>
> Thank you for pointing out the ambiguity! The line 38 was meant for "the first online RGB-only method to achieve dynamic reconstruction from monocular input". We will make this clearer in the camera ready version. In Table S1, DynaMoN is the only existing method to satisfy these conditions. However, as we point out starting from line 9 in the supplementary material, DynaMoN's novel view synthesis performance under larger viewpoint changes remains unvalidated. In contrast, we demonstrate through NVS evaluations that ProDyG achieves dynamic reconstruction with 3D consistency.
>
> > “There are lots of capitalization errors in the References ("3d", "slam", etc).”
>
> Thank you for pointing this out! We will go through the references and make sure to fix these errors in the camera ready version.
>
> ## References
>
> [20] Karaev, N., Makarov, I., Wang, J., Neverova, N., Vedaldi, A., Rupprecht, C.: CoTracker3: Simpler and Better Point Tracking by Pseudo-labelling Real Videos. arXiv preprint arXiv:2410.11831 (2024)
> [54] Sandström, E., Tateno, K., Oechsle, M., Niemeyer, M., Van Gool, L., Oswald, M.R., Tombari, F.: Splat-SLAM: Globally optimized RGB-only SLAM with 3D Gaussians. arXiv preprint arXiv:2405.16544 (2024)

---

> > ### Comment · Reviewer_JEry · 2025-08-04
> >
> > Thank you for the response and for answering my questions. I was also pleased to see the promise of more video comparisons in your response to reviewer KJPL.

---

> > > ### Comment · Area_Chair_hVXW · 2025-08-08
> > >
> > > Dear Reviewer,
> > > Thank you for considering the reply by the authors/other reviews. Please note that you must acknowledge having read the reply and enter your final rating and justification before the deadline. Respectfully, your AC

---

### Official Review · Reviewer_mjPB · 2025-07-03

**Clarity:** 3
**Significance:** 3
**Originality:** 2
**Rating:** 5
**Confidence:** 2

**Summary:**

The paper proposes a method for online dynamic reconstruction from an unposed monocular video stream. The method works by computing masks for moving scene regions in two steps, and using dense bundle adjustment to compute keyframe poses based on the unmasked static scene regions. Interpolated poses for non-keyframes are then used along with scaled depth to generate the static regions of the scene as 3D Gaussians. The dynamic regions are reconstructed and updated using a previous method.

**Questions:**

1. Given that the overhead of optimizing dynamic Gaussians is likely to prevent real-time or even interactive execution, what is the particular use case the authors imagine for an offline method? And doesn't this limit the practicality of the method on the same lines as the previous approach mentioned in the introduction (L26-27)?

2. Since the dynamic reconstruction happens progressively, how does the temporal consistency of the reconstruction compare to the baseline methods? If space allows, it would be insightful to provide some evaluation (either qualitative or quantitative) of the temporal consistency of the proposed approach.

**Ethical Concerns:**

["NO or VERY MINOR ethics concerns only"]

**Final Justification:**

The authors answered my questions in the rebuttal.

**Quality:**

3

**Strengths And Weaknesses:**

Strengths:
1. The task of online 4D reconstruction is novel and has huge potential in practical applications.
2. The paper is well-written, and provides a thorough and detailed overview of the method.
3. Given the constraints of online reconstruction, the results of the paper in comparison to past work (such as SoM) are impressive.

Weaknesses:
1. The method section is quite heavy on details and can become overwhelming. To avoid losing sight of the forest among the trees, I would suggest providing a more intuitive overview of each section, and better motivating the design decisions.

---

> ### Author Rebuttal · Authors · 2025-07-30
>
> Thank you for your input! We respond to the weaknesses and questions you raised as follows:
>
> ## Weaknesses
>
> **1.** Agreed. Thank you for the suggestion! We restructured the method section in the camera ready version to improve readability.
>
> ## Questions
>
> **1.** **We believe that an online method, though not real-time, can be more scalable than a purely offline one.** This is because online systems like ProDyG often work in a sliding window manner rather than processing the entire sequence at once such that the compute demands remain managable even for longer sequences. An online approach also has advantages if more image data is added to improve the quality of a scanned scene without the need to recompute everything from scratch. Our online system is able to reuse previous computations. Also, **our online pipeline shows a paradigm upon which a faster system can be built**. For example, the dynamic Gaussian optimization can potentially be replaced by recent feed-forward methods to approach or mapping optimizations can be made spatially adaptive and sped up with second order solvers to achieve real-time or interactive performance in the future.
> We believe our unoptimized code base can serve as a starting point and as proof-of-concept baseline for future dynamic SLAM methods which may benefit from the many recently proposed speed-ups for both mapping and tracking, but they are out of scope/focus for our work.
>
> **2.** Thank you for the advice! With the Motion Scaffolds being able to warp dynamic Gaussians over time, temporal consistency is considered to be a strength of ProDyG over existing methods. We will add qualitative and quantitative evaluations of point tracking on the iPhone dataset in the camera ready version to demonstrate the temporal consistency of our method. Additionally, we would like to stress that even without the point tracking evaluation, our NVS results shown in Table 2, Figure 4 and the supplementary video are already able to demonstrate how well ProDyG achieves temporal consistency in dynamic regions to a great extent.

---

> > ### Comment · Reviewer_mjPB · 2025-08-05
> > **Maintaining original rating**
> >
> > Thank you for answering my questions.
> >
> > May I ask how you plan to quantitatively evaluate temporal consistency? I don't think the iPhone dataset has ground truth tracks.

---

> > > ### Author Response · Authors · 2025-08-05
> > >
> > > Thank you for your reply!
> > >
> > > The iPhone dataset does in fact include annotations of 5 to 15 "keypoint" tracks per sequence, available in the "keypoint" directory. We plan to quantitatively evaluate both 2D and 3D tracking accuracy by following the procedure used in Shape of Motion [68]. While these keypoints provide sparse rather than dense ground-truth trajectories, they are all located in dynamic regions. As such, we believe this evaluation offers a meaningful measure of temporal consistency.
> > >
> > > References:
> > > [68] Wang, Q., Ye, V., Gao, H., Austin, J., Li, Z., Kanazawa, A.: Shape of Motion: 4D Reconstruction from a Single Video. arXiv preprint arXiv:2407.13764 (2024)

---

> > > > ### Comment · Reviewer_mjPB · 2025-08-06
> > > >
> > > > Thank you for the clarification. I will stick to my original rating of ACCEPT.

---

### Note · Authors · 2025-08-12

We appreciate the constructive reviews and meaningful discussions, which have helped make our work more complete. Below we summarize the outcomes of the discussions with each reviewer:

# Reviewer mjPB

Both questions raised by the reviewer has led to meaningful improvements of our paper: the first prompting us to refine the introduction to **better explain the motivation behind ProDyG being an online yet not real-time system**, and the second suggested an **extra evaluation of temporal consistency**, which we have been working on since the rebuttal.

# Reviewer JEry

The reviewer's questions pointed to some key technical details whose clarification has made our paper easier to follow. Also, we adopted the important advice from the reviewer to include **visualizations of failure modes** on our upcoming project page.

# Reviewer KrLA

The discussion focused on **elaboration on specific components of our method** and the **motivation behind our method design**. We also provided **a detailed runtime profile**, clarifying that **we do not claim real-time performance**. We are glad that our rebuttal have addressed all raised concerns and made our method much clearer to the reviewer.

# Reviewer KJPL

The reviewer insightfully pointed out several potential issues with our work, a significant one being missing ablation studies to justify our seemingly "redundant" method design. In response, we added an important **ablation, demonstrating the necessity of our residual-flow-based segmentation strategy**. We also incorporated a **detailed justification for the method design** into the supplementary material. We are pleased that our rebuttal and discussion have addressed the reviewer’s questions and led to a positive view of the paper.

---

### Decision · Program_Chairs · 2025-09-17

**Decision:**

Accept (poster)

**Comment:**

This paper received in average very positive reviews. Three reviewers gave an accept, while one reviewer gave a borderline reject, however stating that the rebuttal has addressed his concerns. Final review is as of now still outstanding. After having read all the discussions and justifications, the conclusion is that this is a positively received work with only minor issues in the wording as well as an incomplete runtime evaluation or a missing evaluation of temporal consistency. Overall, this AC aligns with the reviewers' view that these problems should be easy to address in the final version, and recommends acceptance for this work.